# Polypharmacy Management in a Gender Perspective: At the Heart of the Problem: Analysis of Major Cardiac Diseases, SARS-CoV-2 Affection and Gender Distribution in a Cohort of Patients in Internal Medicine Ward

**DOI:** 10.3390/ijerph20095711

**Published:** 2023-05-03

**Authors:** Filomena Pietrantonio, Angela Ciamei, Antonio Vinci, Tiziana Ciarambino, Elena Alessi, Matteo Pascucci, Michela Delli Castelli, Silvia Zito, Simona Sanguedolce, Marianna Rainone, Jacopo Di Lorenzo, Fabio Vinci, Giulia Laurelli, Claudia Di Iorio, Roberto Corsi, Serafino Ricci, Alessandra Di Berardino, Matteo Ruggeri, Francesco Rosiello

**Affiliations:** 1Ospedale dei Castelli, Local Health Authority Roma 6, 00041 Albano Laziale, Italy; 2National Centre for Health Technology Assessment, National Health Institute, 00153 Rome, Italy; 3Department of Biomedicine and Prevention, University of Rome “Tor Vergata”, 00133 Rome, Italy; 4Department of Internal Medicine, Luigi Vanvitelli University, 81100 Caserta, Italy; 5Health Management, Local Health Authority Roma 6, 00041 Albano Laziale, Italy; 6Department of Hystological, Anatomical Sciences and Legal Medicine, Sapienza-University of Rome, 00196 Rome, Italy; 7HTA Center, St. Camillus University of Medicine and Health Sciences, 00131 Rome, Italy

**Keywords:** gender medicine, COVID-19, SARS-CoV-2, cardiovascular disease

## Abstract

Background: COVID-19 patients with any pre-existing major cardio-vascular disease (CVD) are at the highest risk of viral infection and of developing severe disease. The pathophysiological mechanism is characterized by the viral link to angiotensin-converting enzyme 2 (ACE2) and the involvement of the endothelial system with the release of cytokines and the inflicting of direct damage to the myocardium, the induction of microthrombosis, and the initiation of alterations in oxygen diffusion. The aim of the study is to analyze the clinical course and outcomes in patients (gender-stratified) with pre-existing major CVD. Methods: Out of the 1833 (973 M/860 F) patients admitted to the Internal Medicine COVID-19 Unit of “Castelli Hospital”, Lazio, Italy, from 1 January 2021 to 31 December 2021, 600 patients (320 M/280 F) with a mean age of 77 (78.6 M/75.1 F) previously had CVD. Demographic characteristics, length of the stay (LOS) and oxygen therapy were evaluated. Results: All of the CVD COVID-19 patients underwent non-invasive ventilation (NIV). CVD was linked with increased LOS (21 days F/22 M) compared to no CVD (19 days). In total, 32.7% of total patients had major CVD. Conclusions: Timely identification and evaluation of patients with pre-existing major CVD are fundamental for adequate treatment based on gender, severity, state of illness and for risk reduction.

## 1. Introduction

Since December 2019, SARS-CoV-2 has been spreading around the world from Wuhan, Hubei Province, China. In 2020, the WHO declared COVID-19 (a SARS-CoV-2-related disease) to be a pandemic [1,2,3,4].

COVID-19 can occur in different clinical forms; severe forms can cause pneumonia, respiratory failure and, in extreme cases, acute respiratory distress syndrome (ARDS).

Patients with comorbidities and those using a large number of medications are the most vulnerable, especially if they have pre-existing cardiovascular disease (CVD) [5,6,7,8].

Other clinical forms may affect the nervous system [9], mental health, [10] or persist beyond the end of the acute infection. These are referred to as “Long COVID” syndromes [11,12,13], another condition that leads to the use of complex therapies. However, in addition to respiratory symptoms, SARS-CoV-2 can trigger an overproduction of proinflammatory cytokine and chemokine storms (tumor necrosis factor-α, interleukin (IL) 1β and IL-6), resulting in multiorgan damage [14].

Underlying CVD is a common condition among hospitalized patients with COVID-19 and is associated with a higher risk of serious illness and mortality. Cardiovascular involvement is often present in critical cases and patients with pre-existing CVD or underlying heart failure appear to be susceptible to myocardial lesions [15,16]. Therefore, it is of fundamental importance to understand the mechanisms that underlie cardiovascular damage from COVID-19.

Several studies have shown that the interaction between the viral spike (S) protein and angiotensin-converting enzyme (ACE) 2, which triggers the entry of the virus into host cells, is likely to be involved in the cardiovascular manifestations of COVID-19 [17].

The damage after SARS-CoV-2 infection may occur at different stages of the disease, and for each stage different physio-pathological mechanisms are involved. Cytotoxic damage is associated with virus infiltration into cells expressing ACE 2 receptors (such as pneumocytes, endothelial cells, cardiomyocytes, and neuronal cells), causing acute lesions in the lungs, vascular system, myocardium, and brain; this usually happens at the early stages of the disease [18,19].

The literature illustrates the role of the renin-angiotensin-aldosterone system (RAAS) in patients, in which SARS-CoV-2 uses ACE 2 to bind to the surface of epithelial cells. However, data on the effect of RAAS inhibitors in COVID-19 patients are not unique. The heart, lungs, kidney, and gastrointestinal tract have a higher expression of ACE 2 which explains systemic and pulmonary hypertension, heart failure, myocardial infarction, and diabetic cardiovascular complications induced by abnormal ACE 2 activity. Potential therapeutic strategies may include preventing the binding of human ACE 2 and SARS-CoV-2 [20].

As the disease progresses, the damage worsens as a result of hypoxemia, local inflammation, and microthrombosis. Furthermore, the increase in cytokines circulation has been demonstrated to be the cause of damage in multiple organs (e.g., stress cardiomyopathy, myocarditis, vasculitis-like syndromes), and systemic inflammation or catecholamine rush is associated with plaque rupture or the blood’s hypercoagulability (thrombotic damage ischemia) [15,21,22].

Subjects with CVD are generally adults, the elderly, and those with several comorbidities. Moreover, age-related immunological quiescence is likely to be associated with severe infection and is considered a predisposing factor [4,23,24,25,26].

Since the beginning of the pandemic, it has been clear that the presence of CVD represents a risk factor that increases the case fatality rate more than any other comorbidity. In a report involving 1591 patients with COVID-19 who were admitted to the intensive care units (ICUs) in Italy, 49% of patients had pre-existing hypertension, 21% had CVD, and 17% had diabetes [27]. Furthermore, in a report of 393 consecutive patients hospitalized with COVID-19 in New York, USA, up to 50% of patients had hypertension (54% of ventilated patients), 36% had obesity (43% of ventilated patients), 25% of patients had diabetes (28% of ventilated patients), and 14% of patients had coronary artery disease (19% of ventilated patients) [28].

On the other hand, both CVD and COVID-19 have a different impact on men and women [8,29].

ASL Roma 6 is a typical Italian Local Healthcare Facility (Azienda Sanitaria Locale, ASL) which is the administrative, commissioning and service provision center for all operations related to public healthcare in Italy under the National Healthcare Service. ASL Rome 6 is representative in terms of both population and complexity (by orogeographical characteristic and hospital territorial integration policies) [30]. “Ospedale dei Castelli” is the only COVID-19 hospital in the entire ASL that hospitalized every COVID-19 patient from all ASL Emergency Rooms.

COVID-19 Internal Medicine Units of the “Ospedale dei Castelli” are also equipped with wireless monitoring systems of vital parameters and electrocardiographic traces. They connect with a portable device (tablet or phone), without the necessity of the constant presence of nursing staff. [31,32,33].

The COVID-19 medical department is organized in such a way to accommodate patients with different intensities of care and different complexities. The number of beds vary according to epidemiological needs, meaning that the department is classified as a “projection department” [4,34].

## 2. Aim of This Study

The primary objective of this study is to investigate the role of gender and pre-existing major CVD in the clinical course of COVID-19 patients that were admitted at the Internal Medicine COVID-19 Units of “Ospedale dei Castelli”. The secondary objective is to produce an economical estimation of related costs.

## 3. Materials and Methods

This study was designed as an observational retrospective analysis of data from clinical documentations of all patients with a complete dataset who were admitted to Internal Medicine COVID-19 Units of “Ospedale dei Castelli” from 1 January 2021 to 31 December 2021.

Major CVD was defined as the occurrence of either:▪Heart failure (HF, a heterogeneous condition in which the heart is unable to pump out sufficient blood to meet the metabolic need of the body);▪Myocardial infarction (MI, necrosis of the myocardium caused by an obstruction of the blood supply to the heart);▪Atrial fibrillation (AF, abnormal cardiac rhythm that is characterized by rapid, uncoordinated firing of electrical impulses in the upper chambers of the heart) and their combination.

Definitions of gender, CVD, HF, MI and AF were derived from by National Institute for Health (NIH), National Library of Medicine, MeSH (last access 4 July 2022).

Admission to COVID-19 units was based on positive molecular SARS-CoV-2 test and chest CT with crazy paving/ground glass aspects.

The following clinical variables were considered:Patient age;Patient gender;Presence of CVD;Presence of other comorbidities, such as:
○Chronic kidney disease (CKD);○Chronic obstructive pulmonary disease (COPD);○Liver disease;○Inflammatory bowel disease (IBD);○Diabetes mellitus (DM);○Connectivitis;○Involutive encephalopathy;○Disthyroidism;

Active neoplasms;WHO pulmonar disease classification [35,36];Clinical outcome, defined as either:
○Discharge○DeathLength of stay (LOS).

Patient study flow is depicted in Figure 1.

The effect summary measure used for the clinical outcome (death vs. discharge) was relative risk ratio (RR). Pearson’s Chi-squared test was used to determine the statistical significance of the results involving the frequencies of categorical variables. Whelch’s *t* test for independent samples (also known as Student’s *t* test for unequal variances) was used for mean comparison in LOS between male and female patients. For all analyses, the significance limit was set at *p* = 0.05.

Economic analysis helped identify LOS as the main driver of the cost estimation together with polypharmacy. Outcome data were compared to both the LOS data and to the intensive care hospitalization data related to recovered and dead patients. The data were obtained from the analytical Instant Report of the COVID-19 organizational model published by the Altems website of the Università Cattolica di Roma [4]. 

With regards to the day of hospitalization value, the estimation used was from the Healthcare Datascience Lab (HD-LAB) from Università Carlo Cattaneo–LIUC di Castellanza in cooperation with Azienda Ospedaliera Nazionale SS. Antonio and Biagio and C. Arrigo, based in Alessandria, and Associazione Ingegneri Gestionali in Sanità worked on this project also. The following cost items have been taken into consideration: human resources involved in the assistance path, devices, equipment and personal protective equipment (PPE) used, lab services, diagnostic services, medicines given to patients, and catering and cleaning services.

## 4. Results

Results are shown in Figure 2 and in Table 1.

In total, 600 CVD patients (320 M, 280 F) and 1233 no-CVD (653 M, 580 F) patients were included in the analysis; overall, 1833 patients were included. In both groups, the average LOS was found to be 1 day greater among men than women (22 vs. 21 days). The number of comorbidities was also greater in men than in women, while no significant difference in age was observed among either group or gender.

The main significant result was found in relation to outcome-related measures. As expected, patients with major CVD were at a higher risk of death than no-CVD patients (RR = 2.89, *p* < 0.001, confidence interval CI: 2.39–3.48), while gender played no role in CVD risk of death. On the other hand, women had a higher risk of death in the no-CVD group than their male counterparts (RR 1.56, *p* = 0.004, CI: 1.15–2.12). Interestingly, no significant difference in outcome was found when analyzing the pneumonia presentation as per the WHO’s classification.

According to the literature [37,38], the total costs of hospitalization in medium-intensity areas, such as COVID-19 Internal Medicine Unit, is EUR 9157/hospitalization/patient, against EUR 22,210.47/hospitalization/patient in high-intensity settings, such as ICU or Intensive Therapy wards. The management of high-intensity CVD COVID-19 patients in COVID-19 Internal Medicine Units with Non Invasive Ventilation (NIV) devices generates a net saving of EUR 13,053.47/hospitalization/patient. By considering only the sample of the 600 CVD COVID-19 patients hospitalized in Internal Medicine COVID-19 Units of “Castelli Hospital”, the savings would amount to EUR 7,832,082. An undisputed economic advantage is generated by managing CVD COVID-19 patients in a COVID-19 Internal Medicine Unit, if it is structured with sub-intensive criteria (NIV and continuous vital signs monitoring) and by transferring to intensive care units only those patients who require invasive ventilation. By managing patients appropriately in a medium-intensity care setting such as a COVID-19 Internal Medicine Unit would lead to savings of over EUR 13,000/hospitalization/patient. 

Future studies could also more accurately investigate potential savings with the use of parametric methods and a quasi-experimental approach.

The presence of one or more CVD increases LOS and, subsequently, inflates costs (every hospitalization day costs EUR 582.38 in medium-intensity care units and EUR 1278.50 in high-intensity ones) [37,39,40,41,42]. On average, male COVID-19 patients stay at least one day more than female patients and, consequently, lead to higher costs. [29] Additionally, it must be pointed out that an increase in LOS also has an impact on the outcome, since the increased length of stay is an independent predictor for complications, morbidity, and negative outcomes in general. Moreover, they are at higher risk of being overloaded with drugs, especially antibiotics, the use of which in Italy has sometimes been subject to misconceptions from both clinicians and patients alike [43].

## 5. Discussion

Major CVDs, in COVID-19 patients, are only common in patients over 70 years old. No gender distribution difference was found among CVD and no-CVD patients.

There were no significant differences in drug therapy and the number of drugs used, which always appear as more than five in both male and female patients affected by COVID-19 and CVD. Likewise, the number of comorbidities did not vary among groups in any significant way. This is probably due to the fact that, due to the current epidemiological transition and the higher survivability of elderly patients, many of the people admitted in Internal Medicine COVID-19 Units have several different clinical diseases which are often not directly related each other.

LOS for COVID-19 patients with combinations of CVD has been reported to be shorter than that for no-CVD patients and this apparent LOS paradox can be easily explained: LOS is shorter for CVD COVID-19 patients because combinations of CVD are important negative prognostic factors and death occurs earlier in such patients, whereas no-CVD patients show a higher rate of recovery. In short, CVD patients simply “fail faster” more often than not.

Higher resource uptake is observed to be directly associated with the increase in the complexity of care, which means an increase in the length of stay in the Unit where the patient is hospitalized, increase in pharmaceutical cost, and also an increase in the reimbursement tariffs that are expected to be associated with single DRGs and, therefore, an increase in the overall public healthcare expenditure. This information can certainly represent a fundamental element that can lay the foundations, from a policy making and health-planning perspective, for a comparison between hospital practices and reference rates for the health activities to be carried out.

Moreover, the increase in resources uptake raises important concerns, particularly with regard to the opportunity costs.

It is generally known that, in a context of limited resources leading to budget constraints, issues related to the effective allocation of resources should be considered as key drivers to pursue productivity gains. In this specific case, the design and the implementation of specific care pathways could lead to the achievement of scale and scope economies, thus also leading to an increase in the quality of care.

It is also important to consider that the appropriate, effective safe and efficient design of care pathways should take into account the correct definition of tasks and responsibilities given to different health professionals. In this case, when possible and consistent with the international guidelines and the Italian laws, task redefinition should be pursued, while also shifting some responsibilities from doctors to specialized nurses. Appropriate training is, of course, required.

In particular, several elements have been outlined, such as overall resources usage; the process critical factors and average LOS, based on the clinical condition of the patient and on the seriousness of the pathology which also implicated a low-intensity cure assistance (characterized by the use of beds in the traditional medical areas without equipment for patients in need of non-invasive ventilation); average complexity (in which all emplacements are designed to manage sub-intensive therapy patients with non-intensive ventilation); and the high cure intensity and aid. 

Throughout this historical period, in order to better manage the pandemic, the evolution and, above all, the clinical course of the disease, assistance structures necessarily had to restructure their processes of diagnosis and treatment, the spaces as a whole, the supply chains, and the related deviations, with a consequent substantial involvement in hospital organization. This has, by necessity, translated into an economic investment, intended for the treatment, management, and treatment of respiratory viral infection from COVID-19. The organizational adjustments put in place to cope with such a high demand were certainly enormous and focused attention on the investments made by the hospital to address the health problem, the supporting technologies normally used in case of epidemics, such as pneumonia with evolution in severe respiratory failure and subsequent assisted pulmonary ventilation, in addition to the prevention of complications such as the use of enteral or parenteral nutrition in cases where the patient is no longer able to feed himself independently. The adoption of these solutions represents an important upgrade in the management of the patient hospitalized in the medical area. The new organizational models developed and, in particular, the management of highly complex patients with the need for non-invasive ventilation in COVID-19 Internal Medicine Units have demonstrated the effectiveness of management in a high intensity medical context, often avoiding transfer to the intensive therapy units with a significant reduction in costs. This model, therefore, appears to be promising and can also be applied to pathologies other than COVID-19, such as acute respiratory failure, acute heart failure, persistent arthritis, stroke, and severe septic states that can find an appropriate location in areas defined as “high complexity”, thus also bringing innovation and development within the departments of Internal Medicine.

## 6. Limitations of the Study

The main limitation of the study is due to its monocentric configuration and, as such, its conclusions may not be as generalizable as those from multicentric studies, although they are in line with the literature and expectations. Moreover, due to differences in people’s behavior during the multiple waves of the pandemic, they do not represent the totality of patients with major CVD who were admitted to the hospital, but only those who were admitted in Internal Medicine wards.

Interference factor and demographic factor adjustment was not performed since no difference in age and gender distribution was found among the disease groups. Likewise, poli-pharmacologic treatment (five or more medications) was equally common among groups.

## 7. Conclusions

The prevalence of major CVD is clinically relevant in patients with COVID-19, particularly in the elderly, and, just as happened with other diseases of the past, long-term consequences should be subject to analyses in the future [20,44]. However, in the short term, timely identification and evaluation of patients with pre-existing major CVD is fundamental for adequate treatment, based on the severity and state of the illness and for risk reduction.

In accordance with the current literature [12], this study shows that there is a strong relation between gender and COVID-19 outcome in patients with pre-existing major CVD, especially in the elderly: during the first three COVID-19 waves, the typical patient in the COVID-19 Unit was old and with three or more comorbidities, treated with more than five drugs. Major CVD was one of the most common co-morbidities for such patients. It is already known that major CVD risk factors are gender related [45] and this study further shows that their presence impacts the LOS and outcome of patients [46]. HF and AF increase LOS since they are correlated with the need for more intensive care, such as NIV and the continuous monitoring of vital parameters.

In addition, prompt identification of such high-risk patients would provide timely treatment and a reduction in even serious complications, with efficacy also on LOS. Therefore, even multidisciplinary management of comorbidities, including the participation of the clinical pharmacist, can be an advantage for the correct treatment of patients with major CVD and COVID-19 diseases. It is an index of appropriateness and effectiveness of the health promotion and care offered, even if prevention and lifestyle rules keep playing an important role. Telemedicine also could perform an important role, enabling physicians to ensure a more appropriate management of complex patients, reducing the time to diagnosis, improving efficiency and efficacy of disease management, and reducing unnecessary clinical visits and hospital admissions. The miniaturized technologies can improve patient adherence. The detection, characterization, and monitoring of major complications also reduces the cost of hospitalization and mortality and assures better quality of life. The use of wireless monitoring systems allows many of these patients to be safely sent home and to effectively integrate with hospital and community services. All this can be integrated with targeted pharmacological strategies, especially in light of recent findings that show how pre-expository prophylaxis and early pharmacological treatment with targeted therapies (such as, antiviral and monoclonal antibodies tailored on the prevalence of a particular serological COVID-19 variant) are effective strategies in reducing the burden of the disease on both the individual and the health system as a whole [47,48].

## Figures and Tables

**Figure 1 ijerph-20-05711-f001:**
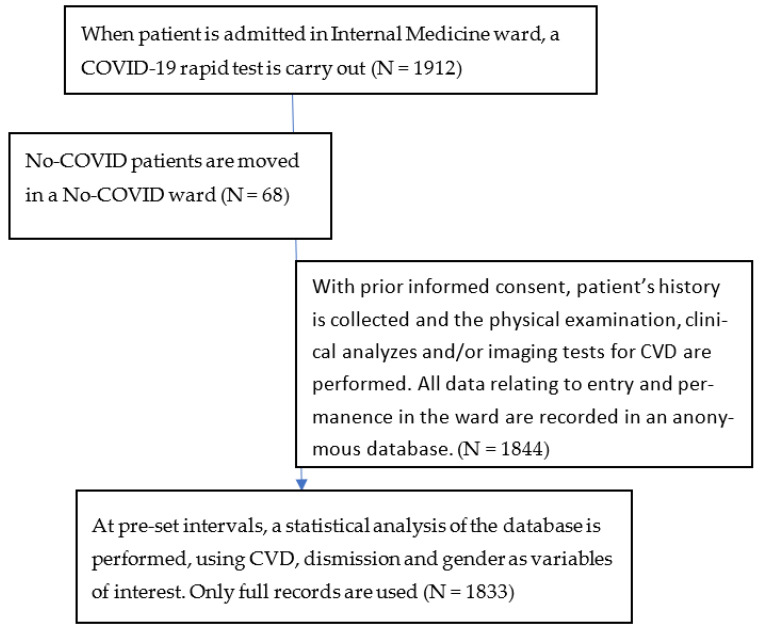
Patients study flow.

**Figure 2 ijerph-20-05711-f002:**
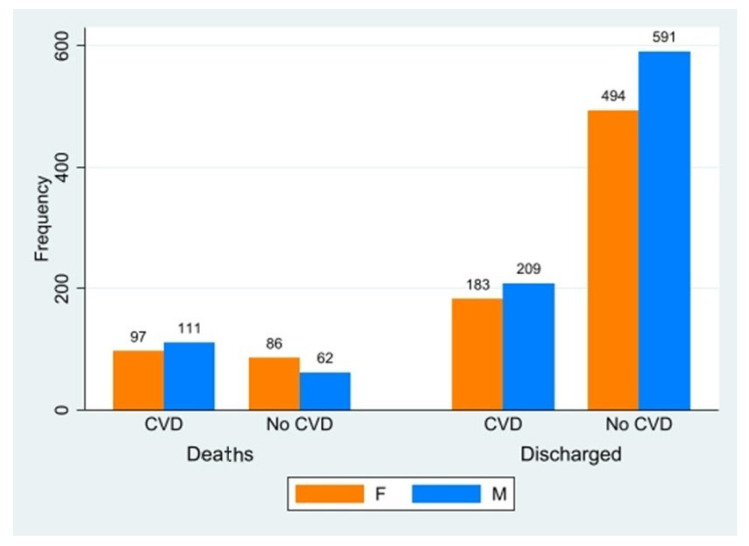
Frequency of deaths and discharged in CVD e No-CVD patients. F: female; M: male.

**Table 1 ijerph-20-05711-t001:** Patient characteristics.

	Deaths	Discharged	Avg. LOS (SD)	Avg. Age (SD)
CVD				
F	97	183	21 (2.2)	78.6 (1.6)
M	111	209	22 (2.1)	75.1 (1.8)
No-CVD				
F	86	494	23 (2.8)	79 (2.0)
M	62	591	23 (2.5)	77.8 (2.2)

CVD: cardiovascular disease; LOS: length of stay. Avg.: average. SD: standard deviation.

## Data Availability

Raw data were generated at PO Castelli Hospital. Derived data supporting the findings of this study are available from the corresponding author F.R. on request.

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
