# Peer review of "Polypharmacy Management in a Gender Perspective: At the Heart of the Problem: Analysis of Major Cardiac Diseases, SARS-CoV-2 Affection and Gender Distribution in a Cohort of Patients in Internal Medicine Ward"

_ijerph, 2023, doi:10.3390/ijerph20095711_

Round 1
Reviewer 1 Report (Previous Reviewer 1)
I thank the authors for sending the revised version.
1. Based on the points mentioned in rebuttal there is supposed to be a Table 1 as well. I can only find the Fig 1. Perhaps, Table 1 got deleted (although it is mentioned in text and the rebuttal document). kindly double check.
2. Parts of the references are now in another language (instead of English). For eg- months are not in English format. Kindly update
Author Response
Dear Reviewer, thank you for your comments. In the uploaded manuscript, we see Table 1 at the end of current figure 2. We hope it is correctly visible now, although we do not know why before it was not so.
Dates and references are now in English format.
Reviewer 2 Report (Previous Reviewer 2)
The study is of very low interest to the clinician, it does notprovide any valuable new data, and does not affect clinical practice.
Even simple arythmetic calculations are incorrect (look at the abstract - first line of the results section - "Out of the 1833 (700 M/599 F) patients "????????)
The exhausting description of the Italian health system is of no value to the paper and the reader
Author Response
Dear Reviewer, thank you for your comments. We believe the added value of the study is the production of clinical and economic data, which can help the organizational decisor in its choices. This is why this paper has been proposed for publication in a public-health oriented journal, rather than a clinical focused one.
We apologize for the incorrect arithmetic calculations; this was due to the fact that, as per editorial recommendations, this is a resubmission of a previous paper, and data was updated during the resubmission.
Introduction text has been reduced in order not to overburden the reader, we hope this change will be satisfactory.
Reviewer 3 Report (New Reviewer)
Since you're using tracked revisions, it's a bit difficult to read. Is it possible to provide a version that does not track revisions for this manuscript?
As for the secondary outcome, the result section did not mention it, and it is recommended to reinforce it by tables or figures.
1. This study did not adjust for related interference factors and demographic factors, and further adjustments are recommended.
2. How is CVD defined or coded? How to confirm that the coding of these patients is correct? Please describe the methodology in detail.
3. Please provide the 95% CI of RR, and present RR in figures.
4. Please present the research process in a flow chart.
Author Response
Thanks for your comments. The newly uploaded version of the paper has been produced without tracking revisions.
The secondary outcome was actually produced, but since it was of not clinical nature, it was misplaced in the Discussion section; we agree this may be confusing, so it has now been moved in Results section, and discussed in proper place.
- Interference factors and demographic factors adjustment was not performed, since no difference in age and gender distribution was found among the disease groups. Likewise, poli-pharmacologic treatment (5 or more medications) was equally common among groups. For this reason, no data adjustment or manipulation was deemed necessary; we added this notion in the Limitation section of the paper.
- We considered only major CVD in this study: Hearth Failure, Miocardial Infarction, and Atrial Fibrillation. This is not uncommon in literature: ex. https://www.mdpi.com/2072-6643/14/23/5172 or https://pubmed.ncbi.nlm.nih.gov/36284679/. The notion of major CVD has now been made more prominent in the text. In case your comment was referred as actual statistical coding, we coded CVD as binary variable, on the basis of the presence of any of the aforementioned three in anamnesis. Also title has been slightly changed to emphasize this.
- CI and measures of dispersion have been explicated in the text.
- Patients Study flow has been added in Methods section, Figure 1.
Round 2
Reviewer 3 Report (New Reviewer)
The revised article can be accepted for publication.
This manuscript is a resubmission of an earlier submission. The following is a list of the peer review reports and author responses from that submission.
Round 1
Reviewer 1 Report
I thank the authors for revising the draft . It is much better than the previous version but still several formatting errors are present. Here are my specific points:
please unify the writing style: in some cases "cardio-vascular" and in some cases "cardiovascular" is written. in many cases CVD is written as abbreviation and again the full form is also used. please make it consistent.
line 246 and 320-331, table 1- are the authors talking about "COVID-19" or just "COVID". please correct / use a uniform writing style
SARS-CoV-2 is an abbreviation so should not be written as "Sars". I do not know why the authors changed it since in the previous version it was written with all capital letters.
line 337: should be "characteristics"
In the previous revision I had suggested to include error bars and SD values. but Now it is unclear what these error bars indicate- SD or SEM? Also a proper figure legend / table legend needs to be added. Also in figure 2 and 3- if the authors have a data set where they are only plotting number of patients per group stratified by gender there is no logic of putting an error bar. In figure 4 with LOS data- adding an error bar is logical.
line 187-221 is better suited in methods section than in introduction.
fig 1 - based on individual numbers mentioned in the pie chart the total number adds upto 288 ( 131+45+26+5+57+19+5); but the authors have mentioned 278 in the text. please check carefully and update as needed.
No information is provided for the "No CVD" group which has 1021 subjects (in table 1 only CVD patients are described) . Are the No-CVD subjects age matched / what is the ratio of male and female candidates in this group? It is nice that in table 2 and 3 the "No CVD" group is already mentioned.
Author Response
1. We unified the writing style using cardio-vascular diseases (CVD)
2. line 246 and 320-331, table 1: we used always “COVID-19" throughout the text.
Table 1 was changed and the new Figure 1 and Table 1 summarize the data contained in the tables of the previous version
3. We used always bthe abbreviation SARS-CoV-2 throughout the text.
4. line 337: should be "characteristics" : it was changed in the new version
5. In the previous revision I had suggested to include error bars and SD values. but Now it is unclear what these error bars indicate- SD or SEM?: Following the current revision of tables and graphs, error bars have not been included. It should be noted that they had been included in the latest version.
6. Also a proper figure legend / table legend needs to be added. Also in figure 2 and 3- if the authors have a data set where they are only plotting number of patients per group stratified by gender there is no logic of putting an error bar. In figure 4 with LOS data- adding an error bar is logical: Following the current revision, figures 2, 3 and 4 have been deleted to improve understanding of the text.
7. line 187-221 is better suited in methods section than in introduction: suggestion was adopted and introduction and methods were changed
8. fig 1 - based on individual numbers mentioned in the pie chart the total number adds upto 288 ( 131+45+26+5+57+19+5); but the authors have mentioned 278 in the text. please check carefully and update as needed: Following the current revision the figures have been summarized in Ffigure 1 and Table 1 currently inserted.
9. No information is provided for the "No CVD" group which has 1021 subjects (in table 1 only CVD patients are described) . Are the No-CVD subjects age matched / what is the ratio of male and female candidates in this group? It is nice that in table 2 and 3 the "No CVD" group is already mentioned: The group of patients without cardio-vascular disease was entered using the data available on the database.
Reviewer 2 Report
I have read the paper several times and could not detect any reasonable logical relation of the title of the trial to the aim of the trial, or to the introduction or discussion or conclusions.
Author Response
We thank the Reviewer 2 for the comments and suggestions and we changed the title of the article linking it to the aim of the trial, the introduction, discussion and conclusions.
We revised English language in the manuscript.
Reviewer 3 Report
The results section is inadequatelly presented, and no statistically analysis is presented
Author Response
We thank the reviewer 3 for the comments and suggestions.
The results section has been improved according to the suggestions of the reviewer and we presented the statistical analysis.